# Discordant Health Implications and Molecular Mechanisms of Vitamin D in Clinical and Preclinical Studies of Prostate Cancer: A Critical Appraisal of the Literature Data

**DOI:** 10.3390/ijms25105286

**Published:** 2024-05-13

**Authors:** Annika Fendler, Carsten Stephan, Bernhard Ralla, Klaus Jung

**Affiliations:** 1Department of Urology, Charité—Universitätsmedizin Berlin, 10117 Berlin, Germany; annika.fendler@charite.de (A.F.); bernhard.ralla@charite.de (B.R.); 2Berlin Institute for Urologic Research, 10115 Berlin, Germany

**Keywords:** prostate cancer, vitamin D, vitamin D status, vitamin D metabolites, 25(OH)D_3_, 1,25(OH)_2_D_3_, calcitriol, clinical studies, preclinical studies

## Abstract

Clinical and preclinical studies have provided conflicting data on the postulated beneficial effects of vitamin D in patients with prostate cancer. In this opinion piece, we discuss reasons for discrepancies between preclinical and clinical vitamin D studies. Different criteria have been used as evidence for the key roles of vitamin D. Clinical studies report integrative cancer outcome criteria such as incidence and mortality in relation to vitamin D status over time. In contrast, preclinical vitamin D studies report molecular and cellular changes resulting from treatment with the biologically active vitamin D metabolite, 1,25-dihydroxyvitamin D_3_ (calcitriol) in tissues. However, these reported changes in preclinical in vitro studies are often the result of treatment with biologically irrelevant high calcitriol concentrations. In typical experiments, the used calcitriol concentrations exceed the calcitriol concentrations in normal and malignant prostate tissue by 100 to 1000 times. This raises reasonable concerns regarding the postulated biological effects and mechanisms of these preclinical vitamin D approaches in relation to clinical relevance. This is not restricted to prostate cancer, as detailed data regarding the tissue-specific concentrations of vitamin D metabolites are currently lacking. The application of unnaturally high concentrations of calcitriol in preclinical studies appears to be a major reason why the results of preclinical in vitro studies hardly match up with outcomes of vitamin D-related clinical studies. Regarding future studies addressing these concerns, we suggest establishing reference ranges of tissue-specific vitamin D metabolites within various cancer entities, carrying out model studies on human cancer cells and patient-derived organoids with biologically relevant calcitriol concentrations, and lastly improving the design of vitamin D clinical trials where results from preclinical studies guide the protocols and endpoints within these trials.

## 1. Introduction: The Aim of This Opinion Paper

Discussions regarding the preventive effectiveness of vitamin D as a nutrient in human skeletal and extra-skeletal health have been ongoing for years. This is evidenced not only through numerous commentaries in social media and daily press articles but also by contributions at annual scientific conferences in which both consensus and controversies have been recorded [1,2,3,4,5,6]. While the impacts of vitamin D on bone diseases such as rickets, osteoporosis, and osteomalacia are well established [7], its roles in extra-skeletal diseases such as cardiovascular, dermatologic, neurological, metabolic, and immunological disorders, and importantly in various cancers remain to be clarified [8,9,10]. In particular, the association between vitamin D status and cancer risk, as well as the potential preventive effect of vitamin D on both the incidence and mortality of cancers, are frequently discussed [11,12,13,14]. Different health effects of vitamin D have been proposed in different cancer entities, necessitating the evaluation of site-specific cancer studies rather than combining data from different cancer entities to estimate health outcomes [13,15]. Therefore, in this opinion paper, we focus on vitamin D in the context of prostate cancer (PCa).

Various observational, randomized controlled, and Mendelian randomized studies—here referred to as clinical studies or trials—have provided conflicting data on the postulated beneficial effects of vitamin D for various PCa outcomes [16,17]. Details of these studies are discussed later (Section 4). The inconsistent results from clinical studies contrast with the findings of experimental preclinical studies on vitamin D [18,19,20,21,22,23,24,25,26,27], which consistently demonstrated cancer-influencing molecular or cellular mechanisms corresponding to vitamin D. This discrepancy between findings in clinical and preclinical studies has been largely ignored within the vitamin D literature to date [28], and is therefore worth further consideration.

To make our concern understandable, we give a brief overview of vitamin D metabolism, the characterization of the vitamin D status through its metabolites, and the molecular effects of vitamin D. Our focus is then critically directed toward preclinical PCa studies, which are often carried out without considering the existing conditions in the human organism concerning relevant vitamin D metabolite concentrations. Subsequently, in connection with the contradictory results obtained from clinical studies, we draw conclusions and formulate propositions for future research.

## 2. Vitamin D Metabolism, Vitamin D Status, and Biological Actions of Vitamin D

### 2.1. Vitamin D Metabolism

The most important steps of vitamin D metabolism are summarized in the simplified vitamin D flow chart shown in Figure 1. For detailed information on the complex metabolism of vitamin D, the reader is referred to current reviews [29,30,31,32]. For clarity, we refer to the essential form of vitamin D for humans, vitamin D_3_, also known as cholecalciferol, in the text below. While 89–90% of vitamin D_3_ is synthesized in the skin from 7-dehydrocholesterol upon exposure to sunlight, only 10–20% of vitamin D_3_ (along with vitamin D_2_ as the other main form of vitamin D, also known as ergocalciferol) derives from dietary intake (Figure 1). After being released into the bloodstream, vitamin D_3_ is transported by vitamin D-binding protein (VDBP) to the liver for hydroxylation to 25-hydroxyvitamin D_3_ (25(OH)D_3_, calcidiol) by the 25-hydroxylases encoded by cytochrome P450 family 2 subfamily R member 1 (*CYP2R1*) and cytochrome P450 family 27 subfamily A member 1 (*CYP27A1*) [33,34]. Megalin, a cell-surface LDL-receptor-related protein 2 (LRP2) in the kidney tubule, binds VDBP and internalizes 25(OH)D_3_, which is subsequently converted to 1,25-dihydroxyvitamin D_3_ (1,25(OH)_2_D_3_, calcitriol) by 1α-hydroxylase encoded by cytochrome P450 family 27 subfamily B member 1 (*CYP27B1*) [35]. This hydroxylation also occurs in extrarenal tissues expressing the vitamin D receptor (VDR), such as the prostate [21,29,36,37]. The hydroxylation step of calcidiol to calcitriol is stimulated by parathyroid hormone, phosphorus, calcium, and fibroblast growth factor 23 with its co-receptor Klotho, but is inhibited by 1,25(OH)_2_D_3_ itself through a feedback mechanism [38,39]. Unlike renal hydroxylase, prostate 1α-hydroxylase has not been found to be regulated by parathyroid hormone and calcium [40]. Both 25(OH)D_3_ and 1,25(OH)_2_D_3_ are degraded by 24-hydroxylase encoded by cytochrome P450 family 24 subfamily A member 1 (*CYP24A1*) into biologically inactive 24,25-dihydroxyvitamin D_3_ (24,25(OH)_2_D_3_) and 1,24,25-trihydroxyvitamin D_3_ (1,24,25(OH)_3_D_3_); see Figure 1.

It is important to emphasize that all four of the abovementioned cytochrome P450 enzymes are also expressed in the prostate [29], such that all enzymes for local synthesis and degradation of vitamin D are available. In prostate cancer, expression of these enzymes as well as *VDR* and *LPR2* is shown to be both increased and decreased compared to normal prostate (Figure 2). In human non-pathological prostate samples, *CYP27B1* expression was found to be higher in young men (<40 years) when compared to elderly men (>60 years), whereas an inverse expression pattern was observed for the *CYP24A1* gene [41]. This decline in the expression of vitamin D-metabolizing enzymes corresponds to an age-related decline in intraprostatic calcitriol levels, indicating that age-dependent loss of the potential protective effects of calcitriol is a potential confounder in clinical studies.

### 2.2. Circulating 25(OH)D_3_ Level as Indicator of the Vitamin D Status

The main circulating vitamin D_3_ metabolites are 25(OH)D_3_, its degraded metabolite 24,25(OH)_2_D_3_, and calcitriol (1,25(OH)_2_D_3_), the functionally active vitamin D metabolite. The use of serum calcitriol as a biomarker for vitamin D status is unsuitable due to the strict control of calcitriol concentration by an autoregulatory feedback system involving parathyroid hormone, fibroblast factor, calcium, and calcitriol itself, as well as its very short half-life [44].

At present, there is a general consensus that circulating 25-hydroxyvitamin D, as the summation of 25(OH)D_3_ and 25(OH)D_2_ with 25(OH)D_3_ as a major vitamin D metabolite, is the most suitable and robust indicator characterizing the individual vitamin D status [4,45,46]. Conclusive observational studies of health outcomes in relation to vitamin D status and randomized controlled trials of vitamin D supplementation require repeated measurements of the vitamin D status on specified dates during the trial [47]. However, the serum concentration thresholds defining vitamin D deficiency status and monitoring the success of subsequent vitamin D supplementation differ quite significantly across various government and scientific society guidelines [28,48]. There is a widespread agreement that <25–30 nmol/L (<10–12 ng/mL) in 25(OH)D concentrations should be considered severely deficient and must be corrected; however, thresholds for optimal 25(OH)D concentrations vary between 50 and 175 nmol/L (20 and 70 ng/mL) in various recent guidelines and recommendations [28,45,48,49,50,51,52,53]. One apparent reason for this dilemma is the lack of harmonization of the various 25(OH)D immunoassays used for measurement in different studies in the past [2,5]. Differences of up to 20 nmol/L (8 ng/mL) between assays were observed within the clinically relevant 25(OH)D range [54]. Therefore, it is unsurprising that not only different healthy vitamin D status thresholds exist, but also that different recommendations for the correction of deficiencies have been suggested in nutritional guidelines [48]. Fortunately, in recent years, there have been many successful efforts that improved comparability between assays by standardizing different commercial test kits using generally accepted reference materials [54,55,56,57,58,59,60].

### 2.3. Calcitriol as Functional Ligand of the Vitamin D Receptor, the Expression of Vitamin D Target Genes, and the Effect on Biological Processes

The circulating calcitriol concentration is sustained in a narrow range by the above-mentioned autoregulatory feedback system, provided that there is no serious vitamin D deficiency (status measured by 25(OH)D concentrations: <25 nmol/L or <10 ng/mL) [61]. The reference ranges of calcitriol in the serum/plasma of healthy adults have been reported to be between 59 and 159 pmol/L, as determined through liquid chromatography with tandem mass spectrometry [62] and between 43 to 228 pmol/L when measured through isotopic and non-isotopic immunoassays [63]. This means that the circulating calcitriol concentration is 1000 times lower than that of circulating 25(OH)D.

Calcitriol is the biologically active vitamin D metabolite in tissues [64,65,66]. As illustrated above (Figure 2), the prostate has a complete set of enzymes that enable the biosynthesis and degradation of calcitriol. The intraprostatic calcitriol level is not a result of the passive diffusion of circulating calcitriol but is predominantly determined by the internalized precursor 25(OH)D_3_ from the circulation and subsequent 1α-hydroxylase activity [44,67].

Within cells, calcitriol acts in the different organs as a ligand of the VDR (recently reviewed in detail in [68,69]; see Figure 1B). Briefly, VDR is a member of the nuclear receptor superfamily, has a structurally conserved ligand-binding domain, and functions as a transcription factor activated by its high-affinity ligand calcitriol [70]. In the ligand binding pocket of the VDR, calcitriol is fixed by three critical pairs of polar amino acids via hydrogen bonds with the three OH groups of calcitriol. The binding of calcitriol changes the protein–protein interaction characteristics of the VDR and its DNA-binding partner retinoid X receptor (RXR). This heterodimeric complex binds to the vitamin D response elements (VDRE). The specific binding of co-activators with chromatin-modifying enzymes is facilitated and increased chromatin accessibility is supported by the rearrangement of nucleosomes at many genomic regions, termed the chromatin model of vitamin D signaling [71]. Because of these epigenome changes, a regulatory loop of the VDR-bound enhancers promoting accessible transcription start sites can be created. Increased or decreased expression of numerous vitamin D target genes leads to the synthesis of corresponding mRNAs, which leave the cell nucleus and are translated into proteins characteristic of various physiological functions [72] (Figure 1B). More than 50 vitamin D target tissues have been identified (https://www.proteinatlas.org/ENSG00000111424-VDR/tissue, accessed on 6 April 2024) and 100 to 500 primary vitamin D target genes can be assumed per tissue [71,73,74]. The different VDR-expressing tissues and cell types are differently equipped with vitamin D target genes [69,75,76]. Vitamin D target genes have been classified into primary target genes, which are directly regulated by the activated VDR, and secondary target genes, which are regulated by factors such as transcription factors that are encoded by primary target genes [72]. Thus, the cell- and tissue-specific pleiotropic effects of vitamin D are mediated through the subsequent cross-talk with tissue- and cell-specific metabolic pathways. Therefore, it is not surprising that Carlberg [72] characterized the biological processes in which vitamin D is involved as follows: “in summary of all VDR-expressing tissues, vitamin D has rather ubiquitous functions”.

## 3. Vitamin D in Preclinical Studies of Prostate Cancer

In the late 1960s, the identification of 25(OH)D_3_ and 1,25(OH)_2_D_3_ as essential metabolites of vitamin D_3_ and the discovery of the VDR created the crucial prerequisites for basic experimental studies exploring the molecular effects triggered by vitamin D in a variety of cancer types [77,78]. Using new advanced technologies in molecular biology, detailed cellular and molecular changes at all omics levels (e.g., genomics, transcriptomics, epigenomics, and metabolomics) that are triggered through the activation of the VDR by calcitriol have now been characterized [21,24,25,79,80]. Finally, the anti-cancer effects of calcitriol have been evidenced in numerous fundamental biological processes, such as cell proliferation, differentiation, apoptosis, cell invasion and metastasis, angiogenesis, epithelial-mesenchymal transition, oxidative stress, and innate and adaptive immunity (recently reviewed in [14,69,81,82,83,84]).

In prostate cancer, the role of biologically active calcitriol—both alone and combined with various drugs—has been investigated in numerous preclinical studies based on cell cultures, complex cell models (e.g., patient-derived organoids), xenografts, ex vivo explants, and animal studies [18,19,20,21,22,23,24,25,26,27]. For example, three different mice models have revealed that mice fed a vitamin D_3_-deficient diet (zero or only 25 IU vitamin D_3_/kg) exhibited significantly faster PCa growth and metastasis and increased severity of prostate intraepithelial neoplastic lesions when compared with mice supplied with a therapeutic vitamin D_3_ diet (1000 or 10,000 IU vitamin D_3_/kg) [23,85,86]. Moreover, using a human PC3 prostate xenograft mouse model, enhanced inhibition of tumor growth was observed when calcitriol was administered in combination with a CYP24A1 inhibitor, which inhibits the degradation of calcitriol by reducing 24-dihydroxylase activity (Figure 1) [87].

In Table 1, the study characteristics of selected cell culture and organoid experiments are compiled, showing the tested calcitriol concentrations and the observed biological effects. The calcitriol concentrations used in these preclinical studies were generally between 10 and 100 nmol/L; however, the calcitriol concentrations used in the experiments should correspond to those determined in human prostate tissue before and after different doses of vitamin D supplementation. Meaningful experimental results on the biological effects of vitamin D can only be expected if the calcitriol concentrations used in the experiments are comparable to those present in the prostate.

To the best of our knowledge, only two studies to date have analyzed vitamin D metabolites in human prostate tissue samples [37,97]. Wagner et al. [97] analyzed the vitamin D metabolite levels of 25(OH)D_3_ and 1,25(OH)_2_D_3_ in serum and tissue samples from four patient groups after radical prostatectomy in a randomized clinical trial. Patients in the control arm were without vitamin D supplementation, while treated patients received 400 IU, 10,000 IU, or 40,000 IU of vitamin D_3_ per day for 3 to 8 weeks before surgery. Serum and prostate tissue levels of 25(OH)D_3_ and 1,25(OH)_2_D_3_ increased in a dose-dependent manner and were highly correlated with and among each other. However, regression analyses showed positive intercepts for both metabolites in prostate samples at corresponding extrapolated zero serum concentrations. These results confirmed that there was a basal intraprostatic vitamin D metabolism that was not determined by the corresponding serum concentration [97]. PCa tissue samples from different prostate zones showed comparable calcitriol levels of 25 to 35 pmol/kg. It is worth noting that, even in patients with a vitamin D dosage of 10,000 IU or an extremely high dosage of 40,000 IU per day over weeks, the calcitriol levels in the prostate did not exceed 50 and 80 pmol/kg, respectively. In these patients, individual serum levels of calcitriol were between 60 and 190 pmol/L and 100–210 pmol/L, respectively, corresponding to the abovementioned upper reference range for serum calcitriol.

In the second study, Richards et al. [37] analyzed benign prostate tissue samples from African and European American PCa patients who underwent radical prostatectomy without additional vitamin D supplementation. The calcitriol levels (95% confidence intervals) were 65–96 pmol/kg for European American patients and 43–67 pmol/kg for African American patients. The well-concordant values of both studies support the conclusion that calcitriol levels in benign and malignant prostate tissue samples are below 100 pmol/kg. Considering the reference units of liter and kg to be largely equivalent, this means that the calcitriol concentrations used in all of the above-mentioned preclinical studies (Table 1) exceeded the intraprostatic concentrations several times, usually by a factor ranging from 100 to 1000.

## 4. Vitamin D in Clinical Studies of Prostate Cancer

Both observational studies and randomized controlled clinical trials (RCT) have been conducted to validate the effects of vitamin D on human health. In oncological clinical studies, cancer-typical factors such as incidence, recurrence, progression, survival rate, and mortality can serve as target criteria (endpoint, outcome) for estimating the potential health benefits of vitamin D. These outcome parameters are examined in observational studies in relation to baseline serum 25(OH)D concentrations, as the accepted indicator of the vitamin D status, and in RCTs regarding intervention-type research involving vitamin D supplementation. In the following, we refer to data from meta-analyses, which enable a comprehensive assessment of the research question by combining the effect sizes of several quality-tested studies.

Since the 1990s, numerous observational studies—mostly case-control studies, nested case-control studies, and both retrospective and prospective studies—have been published on the incidence risk of PCa. In 2011, Gilbert et al. [98] illustrated the heterogeneity of the various vitamin D studies, which did not allow for the association between vitamin D status and PCa risk to be conclusively proven. In 2014, Xu et al. [99] conducted a meta-analysis based on 21 vitamin D studies involving 11,941 cases and 13,870 control participants. Men with higher serum 25(OH)D_3_ concentrations were found to have a significantly increased risk of PCa by 17%. In 2018, another meta-analysis was carried out by Gao et al. [100], without having taken note of the study by Xu et al. [99]. Gao’s meta-analysis was based on 19 studies, including only nested case-control studies and prospective cohort studies. However, the findings of Xu et al. [99] were confirmed in that higher serum 25(OH)D_3_ levels were significantly associated with an increased PCa risk (relative risk, RR: 1.15; 95% CI: 1.06–1.24), accompanied by a dose–response effect by 25 nmol/L (10 ng/mL) 25(OH)D_3_.

Travis et al. [101] conducted a collaborative analysis of individual data from 19 prospective studies to assess the relationship between serum 25(OH)D_3_ and PCa risk. This meta-analysis included 13,462 men with incident PCa of different degrees of aggressiveness and 20,261 men in the control arm. Of these, 14 and 12 studies were identical to those analyzed by Gao et al. [100] and Xu et al. [99], respectively. The serum 25(OH)D_3_ concentration was positively associated with total PCa risk (odds ratio, OR: 1.22; 95% CI: 1.13–1.31) but varied depending on the degree of PCa aggressiveness. As shown in a detailed sub-analysis, higher serum 25(OH)D_3_ was related to non-aggressive PCa (OR: 1.24; 95% CI: 1.13–1.36), but not to aggressive PCa (OR: 0.95; 95% CI: 0.78–1.15) [100].

Conversely, no association was reported between the baseline serum 25(OH)D_3_ level and PCa risk in a study including 4065 men with PCa from the Danish PCa Registry [102]. In the Kuopio Ischaemic Heart Disease Risk Factor Study, the pre-diagnostic 25(OH)D_3_ level and PCa incidence were evaluated in 2578 men with a follow-up time of 35 years and adjusted for other risk variables [103]. Similarly, a connection between vitamin D status and PCa incidence was not detected.

In contrast to these PCa risk results, there have been reports on PCa-specific mortality, progression, and recurrence in relation to serum 25(OH)D_3_ levels [13,17,104,105,106,107,108]. A dose–response meta-analysis of seven cohort studies showed that the mortality significantly decreased with higher 25(OH)D_3_ levels, with an increment for every 20 nmol/L (8 ng/mL) (hazard ratio, HR: 0.91; 95% CI: 0.87–0.97) [17]. In a large biobank cohort of 193,842 men, PCa patients with deficient baseline serum 25(OH)D_3_ (<30 nmol/L or <12 ng/mL) levels experienced higher mortality (HR: 1.36; 95% CI: 1.06–1.75) than patients with sufficient 25(OH)D_3_ levels (>50 nmol/L or >20 ng/mL) [13].

Other studies have reported that higher serum 25(OH)D_3_ levels were associated with a 57% reduced risk of lethal PCa or an improved prognosis [104,105,106]. In a study including 3849 patients who underwent radical prostatectomy, the pre-operative serum 25(OH)D_3_ concentrations were not associated with histological tumor grade, pathological tumor stage, or biochemical recurrence [107]. This finding was supported by the authors in a prospective prostate biopsy study including 480 men with suspected PCa, of whom 222 had prostate cancer and 258 had no evidence of malignancy [108]. The mean serum 25(OH)D_3_ concentrations did not differ between men with PCa and men with no evidence of malignancy and were not related to Gleason grade as a progression indicator. These exemplary observational studies highlight the contradictory results found between vitamin D status and outcome parameters not only in different studies but also in different meta-analyses.

Several RCTs have been conducted to evaluate the potential benefits of vitamin D supplementation on pancancer incidence and mortality, but few appropriate studies have presented data on site-specific cancers. Therefore, information about the results of pancancer outcomes in relation to vitamin D supplementation and meta-analyses are considered here to assess the situation for PCa. The first comprehensive pancancer meta-analysis, which was based on thirty RCTs with vitamin D alone and analyzed publications from between 1945 and 2017, was carried out in 2018 [109]. Goulao et al. [109] did not find evidence that cancer incidence and cancer mortality were reduced with vitamin D supplementation, independently from its dosage and type of supplement (cholecalciferol, ergocalciferol). An updated meta-analysis by Keum et al. [110], which included studies using higher doses of vitamin D, confirmed the failed relationship between vitamin D and pancancer incidence (RR: 0.98; 95% CI: 0.93–1.03), but revealed an improved pancancer survival rate (RR: 0.87; 95% CI: 0.79–0.96). In the large 5.3-year VITAL trial, 12,927 and 12,944 participants received a daily 2000 IU vitamin D_3_ medication or a placebo, respectively [12,111]. Vitamin D supplementation did not significantly reduce the primary outcome of pancancer incidence (HR: 0.96; 95% CI: 0.88–1.06) or the second endpoint of PCa incidence (RR: 0.98; 95% CI: 0.72–1.07). The most recent meta-analysis from 2023, which included 14 RCTs published between 2003 and 2022, found non-significant effects of vitamin D on pancancer mortality (RR: 0.94; 95% CI: 0.86–1.02) and on PCa mortality (HR: 0.30; 95% CI: 0.08–1.07) [112]. This result was confirmed through an additional meta-analysis involving the individual patient data of the trials used in the conventional meta-analysis. However, sub-analysis of the 10 studies with daily vitamin D administration showed a 12% lower cancer mortality in the vitamin D group compared to the placebo group, but not in the remaining 4 studies using a bolus regimen.

Several recent reviews have critically analyzed the possible reasons for the inconclusive and conflicting outcome data of these two types of vitamin D-related clinical trials and, likewise, between the two studies [16,83,113,114,115]. In addition to the analytical issues mentioned above, typical shortcomings of the design of these clinical studies include the number and selection of study participants, different follow-up times, different thresholds for vitamin D status, unconsidered confounding variables, and missing/insufficient adjustments of other influencing variables. Except the typical flaws in clinical studies, vitamin D-specific seasonality must be particularly considered when evaluating vitamin D studies. Stamp and Round [116] were among the first authors to report seasonal changes in serum 25(OH)D concentrations with significantly lower values in winter/spring compared with values in summer/autumn. They attributed these results from a two-year follow-up study to increased vitamin D_3_ synthesis in the skin due to varying levels of sun exposure. This is now a well-recognized fact [117] and must be considered in the context of climatic, geographical (latitudes), and environmental (air pollution) factors that influence the solar intensity for sufficient vitamin D_3_ synthesis and can alter the 25(OH)D_3_ concentrations [118,119]. Statistical evaluation can also lead to incorrect conclusions, as shown by a very comprehensive vitamin D meta-analysis with 500,962 participants that was published in 2021 which recently had to be corrected [10].

Furthermore, for a successful vitamin D-related study, it is particularly important to consider specific characteristics of nutrient-related studies. Heaney [120] summarized these special features and requirements through a guideline and defined rules for such studies; in particular, for vitamin D, they include (a) the measurement of vitamin D status at baseline and repeated measurements to verify vitamin D status in the follow-up, (b) the consideration of vitamin D deficiency as an inclusion criterion, and (c) the application of a meaningful intervention that is capable of altering the vitamin D status. However, it should be understood that following these requirements in clinical trials is particularly challenging, as detected vitamin deficiencies require correction and cannot be pursued in the long term from an ethical point of view [113]. All of these pitfalls explain the inconsistencies between vitamin D studies in PCa patients, as mentioned above.

## 5. Discordance between Preclinical and Clinical Vitamin D Related Study Results in Prostate Cancer

As discussed above in Section 3, the calcitriol concentrations used in preclinical studies are generally 100 to 1000 times higher than the intraprostatic concentrations found under both normal 25(OH)D_3_ status and with vitamin D supplementation [37,97]. Therefore, it may be misleading to consider 100-fold higher calcitriol concentrations of 10 nmol/L in model experiments as biologically relevant concentrations [25]. This is also illustrated by the fact that the circulating levels of calcitriol and 25(OH)D_3_ are highly correlated with the corresponding 30 to 70% lower intraprostatic tissue levels in the same picomolar concentration range (Spearman’s rank correlation coefficients > 0.73, *p* = 0.0001) [97]. Moreover, all study approaches based on calcitriol supplementation with extremely high peak serum levels of 978–2420 pmol/L and even 7–11 nmol/L (partly in combination with chemotherapeutics) in metastatic PCa patients were not successful in improving outcomes [121,122,123]. Performing calcitriol studies in PCa patients undergoing active surveillance has been suggested [124]. However, there are no data on the intraprostatic concentrations of vitamin D metabolites at such high serum concentrations. Similar to the androgen saturation model postulated for PCa [125], it cannot be excluded that intraprostatic calcitriol homeostasis is protected from serum changes and extremely high serum levels, as local calcitriol concentration is determined primarily through local synthesis of calcitriol from the uptaken serum 25(OH)D_3_.

Therefore, we believe that the use of biologically irrelevant high concentrations of calcitriol in preclinical studies raises reasonable concerns regarding the postulated biological relevance of the “myriad of biological effects” [68] and numerous subsequent biological processes [14] observed in such experiments. This appears to be true not only for PCa but also for other cancer entities, as calcitriol concentrations in other tissues are comparable to those in the prostate (e.g., colon tissue with a mean value of 28 pmol/kg [126]).

It is reasonable to assume that the molecular VDR-mediated changes triggered by extremely high calcitriol concentrations in experimental studies do not reflect the true biological situation, both quantitatively and especially qualitatively. Treatment of breast cancer tissue slices with 0.5 nmol/L or 100 nmol/L of calcitriol and analysis with an Affymetrix hybridization microarray showed clear dose–response-dependent transcriptional effects [127]. When compared to non-treated controls, 9 and 196 differentially expressed transcripts were found at 0.5 nmol/L and 100 nmol/L, respectively [127].

Within the PCa tissue samples in the abovementioned randomized clinical trial by Wagner et al. [97], the expression of *CYP24A1*—as a characteristic indicator of VDR and calcitriol function—did not statistically differ between the subdivided samples regarding tertile calcitriol concentrations in the ranges of 15–24, 25–33, and 34–79 pmol/kg [21]. In contrast, in patient-derived benign prostate epithelial organoids using 10 nmol/L calcitriol and subsequent single-cell RNA sequencing analysis, increased *CYP24A1* mRNA was found together with numerous transcripts of other pathways [25]. These data underscore the dose–response effects of calcitriol that need to be considered in the future design of experimental studies to determine the actual effects of calcitriol at biologically relevant concentrations. Aside from direct tissue-specific effects, vitamin D also modulates immune responses and inflammation in cancer [128,129], an effect that is not accountable for in cell line or xenograft models and may contribute to the observed discrepancies between preclinical and clinical studies.

The discrepancies between preclinical and clinical studies highlight that preclinical outcomes result in well-documented and biologically explainable changes at the molecular and cellular levels, while clinical studies show inconsistent and conflicting results regarding integrative outcome parameters such as incidence, progression, and mortality. This suggests that preclinical experimental designs with non-biologically relevant high calcitriol concentrations are a major reason for this discordance.

## 6. Conclusions and Outlook for Future Research

The literature data on the effects of vitamin D in the context of prostate cancer reveal clear differences between results from preclinical and clinical studies. We have outlined here that while preclinical studies demonstrated pronounced effects of vitamin D treatments at molecular and cellular levels, these are not translated to consistent effects on clinical outcomes including PCa incidence and mortality within clinical trials. A key reason for this discordance seems to be the higher concentrations of calcitriol used in preclinical studies which are not found naturally in human prostate. We believe that this discordance is transferable to other cancer entities, although detailed data on tissue concentrations of vitamin D metabolites are currently lacking.

To address this disparity and avoid misleading conclusions, future research should: (a) collect detailed metabolite profiles of vitamin D for the various target organs of interest under physiological and pathological (cancer) conditions, (b) conduct well-designed studies using corresponding biologically relevant calcitriol concentrations in cancer or stem cells derived from human organs and in patient-derived organoids, and (c) use the results to improve the design of vitamin D clinical trials through meaningful and feasible molecular biology analyses [130,131].

## Figures and Tables

**Figure 1 ijms-25-05286-f001:**
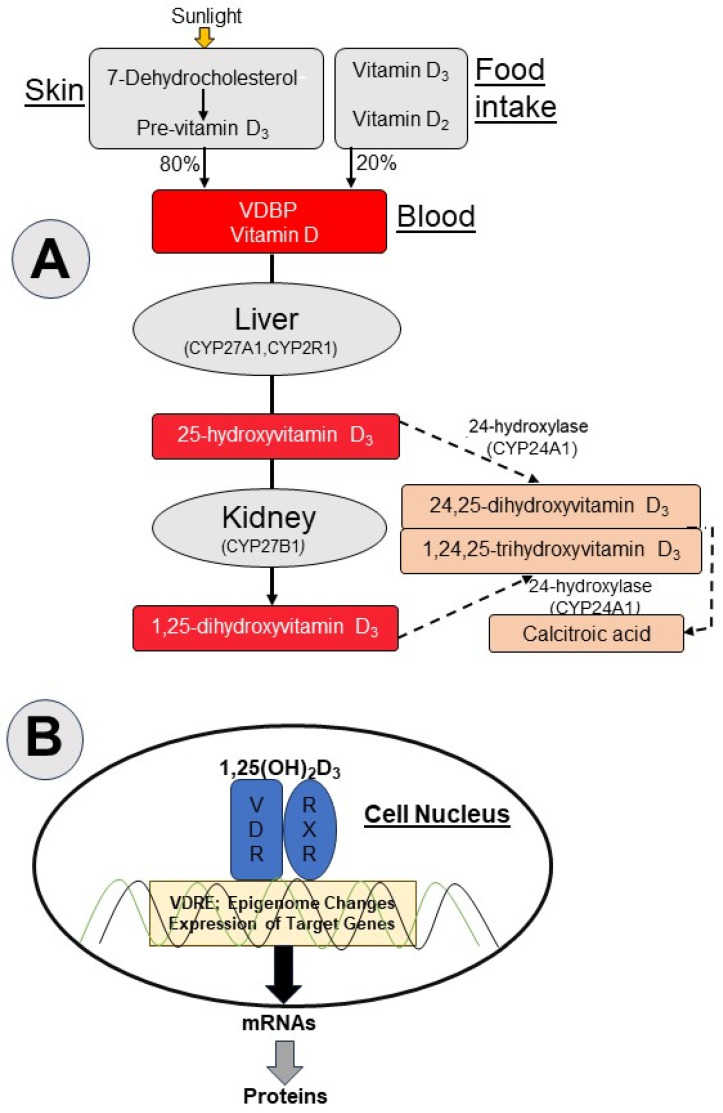
Simplified schemes of (**A**) biosynthesis and degradation of vitamin D and (**B**) the calcitriol action as ligand of the vitamin D receptor and the subsequent epigenome and transcriptome changes. The metabolizing vitamin D hydroxylases are members of the cytochrome P450 family and are indicated abbreviated in parentheses. Abbreviations for subfigure (**A**): Vitamin D binding protein (VDBP); 25-hydroxylases encoded by cytochrome P450 family 2 subfamily R member 1 (*CYP2R1*) and cytochrome P450 family 27 subfamily A member 1 (*CYP27A1*); 1α-hydroxylase encoded by cytochrome P450 family 27 subfamily B member 1 (*CYP27B1*); 24-hydroxylase encoded by cytochrome P450 family 24 subfamily A member 1 (*CYP24A1*). (**B**) Vitamin D receptor (VDR); Retinoid X receptor (RXR); vitamin D response elements (VDREs). The different colored lines symbolize the two DNA strands.

**Figure 2 ijms-25-05286-f002:**
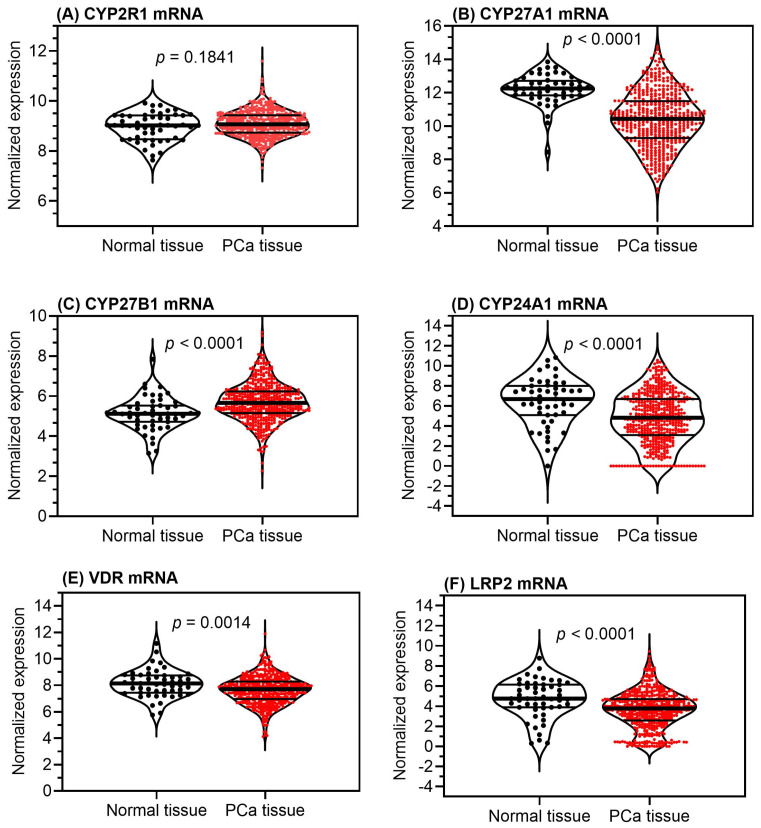
Gene expression of the metabolizing vitamin D hydroxylases of the cytochrome P450 family, the vitamin D receptor, and the LDL-receptor-related protein 2 in tissue samples of normal prostate and prostate cancer. Data from The Cancer Genome Atlas (TCGA) downloaded from UCSC Xena (https://xena.ucsc.edu/, accessed on 28 March 2024) were used [42,43]. Further details are given in the Appendix A. Abbreviations: Cytochrome P450 family 2 subfamily R member 1 (*CYP2R1*); cytochrome P450 family 27 subfamily A member 1 (*CYP27A1*); cytochrome P450 family 27 subfamily B member 1 (*CYP27B1*); cytochrome P450 family 24 subfamily A member 1 (*CYP24A1*); vitamin D receptor (*VDR*), LDL-receptor-related protein 2 (*LRP2*).

**Table 1 ijms-25-05286-t001:** Cellular, transcriptional, and metabolic effects in experimental calcitriol treatment studies.

Reference, Year	Study Object ^a^	Calcitriol (nM) Treatment	Effects of Calcitriol
Blutt et al., [88] 2000	LNCaP cells	10–100	Cell growth inhibition Enhanced apoptosis
Krishnan et al., [89] 2004	LNCaP cells	50	Numerous up-and down-regulated genes
Stewart et al., [90] 2005	LNCaP, DU 145, PC-3, C4-2, LAPC-4 cells	100	Cell growth inhibition independent on IGFBP3
Bao et al., [18] 2006	LNCaP, PC-3, DU 145 cells	100	Decreased PCa cell invasion by TIMP1 modulated MMP9
Bao et al., [91] 2008	DU 145, BPH-1, RWPE-1, CWR22R cells	100	Antioxidative effects in non-malignant prostate cells through G6PD activation
Kovalenko et al., [92] 2010	RWPE-1 cells	100	Multiple calcitriol-regulated tran-scripts of anticancerogenic path-ways (WNT, Notch, NFKB1)
Hidalgo et al., [93] 2011	Benign- and cancer-associated prostate fibroblasts	100	Cell-dependent VDR-mediated transcriptional activities
Giangreco et al., [94] 2013	RWPE-1, RWPE-2, LNCaP, primary cells isolated from human prostate	10–100	Upregulation of tumor suppressive miRNAs (miR-100, miR-125b)
Singh et al., [79] 2015	RWPE-1, RWPE-2, HPr1, HPr1AR, LNCaP, C4-2, PC-3 cells	100	Identification of VDR-regulated miRNA patterns and patterns of miRNA and mRNA co-regulation
Abu El Maaty et al., [22] 2017	LNCaP, VCaP, DU 145, PC-3 cells	100	Disruption of glucose metabolism and the tricarboxylic acid cycle
McCray et al., [25] 2021	Patient-derived benign prostate epithelial organoids	10–50	Inhibition of WNT activity and suppression of DKK3
Erzurumlu et al. [95] 2023	LNCaP, 22Rv1 cells	2.5–100	Inhibition of the androgen receptor signaling and tumor formation of LNCaP cells

^a^ Applied cell lines according to the nomenclature suggested by Germain et al. [96]: cell lines derived from normal prostate epithelial cells (BPH-1, RWPE-1, HPr1, HPr1AR); cell lines derived from PCa as androgen-dependent PCa cell lines (LNCaP, LAPC-4, VCaP) and androgen-independent cell lines (PC-3, DU 145, C4-2), and androgen-sensitive cell lines (CWR22R) that can proliferate without androgens, but show a faster proliferation if androgen is a component in the culture medium. Abbreviations: Insulin-like growth factor binding protein 3 (IGFBP3); prostate cancer (PCa); TIMP metallopeptidase inhibitor 1 (TIMP1); matrix metallopeptidase 9 (MMP9); glucose-6-phosphate dehyrogenase (G6PD); nuclear factor kappa B subunit 1 (NFKB1); vitamin D receptor (VDR); microRNA (miR); dickkopf WNT signaling pathway inhibitor 3 (DKK3).

## Data Availability

Discussed data are given in the references and in the Appendix A.

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
