# Peer review of "Discordant Health Implications and Molecular Mechanisms of Vitamin D in Clinical and Preclinical Studies of Prostate Cancer: A Critical Appraisal of the Literature Data"

_ijms, 2024, doi:10.3390/ijms25105286_

Round 1

Reviewer 1 Report (Previous Reviewer 2)

Comments and Suggestions for Authors

All prior concerns have been addressed.

Author Response

RESPONSES TO REVIEWER 1 Comments

We thank the reviewer for taking the time to re-examine our revised manuscript and are pleased that we were able to address the previous concerns.

Reviewer 2 Report (New Reviewer)

Comments and Suggestions for Authors

Manuscript ID: ijms-2995623

Title: Discordant Health Implications and Molecular Mechanisms of Vitamin D in Clinical and Preclinical Studies of Prostate Cancer: A Critical Appraisal of Literature Data

Authors: Annika Fendler et al.

The manuscript is well written and presents in critical and rational way the reasons for discrepancies in the effects of vitamin D use in prostate cancer between preclinical studies and clinical observations. Of particular importance are the main imperfections of the research conducted on cell lines presented by the authors, which make the clinical usefulness of these tests negligible. On the other hand, the manuscript some minor inaccuracies that should be removed.

  1. Abstract, line 18. Authors should consider replacing “active vitamin D metabolite vitamin D metabolite 1,25-dihydroxyvitamin D3 (calcitriol) in tissues” with “active vitamin D metabolite, 1,25-dihydroxyvitamin D3 (calcitriol) in tissues”
  2. Abstract, line 26. The statement relates to results from multiple preclinical studies, not a single study plus the comment 2. For this reason, “preclinical study results” should be replaced with “the results of preclinical in vitro studies”.
  3. Line 78. Abbreviation “(LRP2”. All abbreviations should be presented in their full name at the point where they appear for the first time, starting from the abstract. Full names of abbreviation should be repeated in the Introduction, Discission, and each figure legend, table header or table caption.
  4. Line 82-83. “The calcitriol hydroxylation step is stimulated by parathyroid hormone” is unclear and misleading and should be replaced with “The hydroxylation step of calcidiol to calcitriol is stimulated by parathyroid hormone”.
  5. Line 127-128. The authors stated that “At present, there is a general consensus that circulating 25(OH)D3 is the most suitable and robust indicator characterizing the individual vitamin D status”. Whereas, in lines 134-137, authors stated that “There is a widespread agreement that <25-30 nmol/L in vitamin D concentrations should be considered severely deficient and must be corrected; however, thresholds for optimal vitamin D concentrations vary between 50 and 175 nmol/L in various recent guidelines and recommendations”. Does the latter statement refer to 25(OD)D3 or vitamin D3? If it concerns vitamin D3, a comment is necessary in the text of manuscript. In addition, the authors should present concentration in nmol/L, as well as in ng/ml because both type of units is used in literature (PMID: 31101452).
  6. Lines 205-208. “mice supplied with a sufficient vitamin D3 diet (1000 or 10,000 IU vitamin D3/kg)” Doses of vitamin D3 of 1,000 or 10,000 IU/kg should not be called sufficient, but therapeutic or supranormal.
  7. In the chapter “Discordance between Preclinical and Clinical Vitamin D Related Study Results in Prostate Cancer”, or in Conclusions, the author should write that preclinical studies are conducted on cancer sell lines, or in animal models in which cancer cells are transplanted into animals’ bodies. The use of xenogeneic cancer cells in animal models means that possible limitation in the progression of prostate cancer may results from the anticancer effects of the tested compounds, as well as the increased effectiveness of the immune system in combating antigenically incompatible cells. For these reasons, preclinical research may lead to new research directions, but the final validation of therapeutic concepts must be performed in the clinic.
Comments on the Quality of English Language

se commens for authors

Author Response

Reviewer 3 Report (New Reviewer)

Comments and Suggestions for Authors

In the manuscript submitted to me for review entitled " Discordant Health Implications and Molecular Mechanisms of Vitamin D in Clinical and Preclinical Studies of Prostate Cancer: A Critical Appraisal of Literature Datathe authors Annika Fendler, Carsten Stephan, Bernhard Ralla and Klaus Jung discuss issues related to discrepancies between conclusions reached in clinical and preclinical studies on the effect of vitamin D in prostate cancer patients.

The research has been carried out extremely thoroughly, presenting the accumulated data on the subject from research by other authors over a period of time longer than half a century. This shows how important the topic discussed in the present manuscript is to human health. Collecting, collating, and summarizing so much information, even if it is conflicting in many of the results presented, may help future research uncover the mechanisms by which vitamin D may contribute to improved health status and lead to greater percent survival in patients with prostate cancer.

In their study, the authors used 124 references, of which nearly 1/2 are from the last 5 years, which shows that the topic has been actively worked on by various research teams in recent years, and the summary data presented in this manuscript would attract the attention of readers of IJMS. The included data is summarized using 2 well-designed and presented figures and 1 table.

My remarks and recommendations to the authors are:

As I mentioned above, the research was done in great detail covering different points of view of the problem. My only concern is with the Conclusion section. The last passage of section "5. Discordance between Preclinical and Clinical Vitamin D Related Study Results in Prostate Cancer" and the first paragraph of "Conclusion and Outlook for Future Research" overlap to a large extent. In the second paragraph of the Conclusion, in my opinion, the personal opinion of the authors is expressed, which would affect a large part of the readers. It is true that the manuscript itself represents Opinion, but I think that this opinion should be expressed as a conclusion not so sharply, because presented in the current version the statement that such: "unsuitable experimental approaches and the questionable conclusions" from the preclinical studies casts doubt on the competence of the cited authors who presented the results of experiments in vitro. The authors of the present manuscript wanted to emphasize the fact that in preclinical studies the results are always more positive than in clinical application, but this is a well-known fact. When applying a therapeutic agent to an organism, many factors such as the reaction of the immune system, the type of diet applied and many others are included, which can change the course of the therapy, which do not exist in in vitro experiments.

My suggestion is that at the end of the well-presented results in the previous points of the manuscript, the authors should rewrite the conclusion so that it does not sound so critical.

Round 2

Reviewer 2 Report (New Reviewer)

Comments and Suggestions for Authors

Manuscript ID: ijms-2995623

Title: Discordant Health Implications and Molecular Mechanisms of Vitamin D in Clinical and Preclinical Studies of Prostate Cancer: A Critical Appraisal of Literature Data

Authors: Annika Fendler et al.

The new version of the manuscript does not raise any objections and, in the reviewer’s opinion, is practically ready for publication. However, after re-reading the manuscript, the reviewer found that the authors could consider discussing the relationship between vitamin D levels in patients and the period over which these levels are measured (PMID: 32188088). This relationship may cause significant differences in vitamin D concentration and therefore, the authors should suggest that, in addition to determining the vitamin D level, the time of year in which the tests were performed should also be recorded. The results should be collected and analyzed separately for summer and winter periods.

Author Response

This manuscript is a resubmission of an earlier submission. The following is a list of the peer review reports and author responses from that submission.

Round 1

Reviewer 1 Report

Comments and Suggestions for Authors

This opinion of authors is presented in a very analytical form. 

Reviewer 2 Report

Comments and Suggestions for Authors

This is an opinion manuscript regarding discrepancies in the conclusions of preclinical and clinical studies about the benefits of vitamin D in Prostate cancer. However, this manuscript required some improvement in the organization of the ideas and paragraphs:

1. In order to understand the discrepancies between preclinical and clinical studies, the manuscripts would benefit if the introduction began with an explanation of vitamin D biology: synthesis, degradation, effects of the active form, Calcitriol on cell signaling activities through the binding to VDR. So, when the reader reads about calcitriol concentration or VDR will understand what the authors are talking about.

2. After the introduction, the logical steps would be talking about the preclinical data and their findings, as well as specific examples; saying that Calcitriol induces epigenetic, transcriptome, and proteomic changes does not say anything to the reader. What is the meaning of those changes? What is the benefit? Giving a few examples would improve the message. Also, the authors talk about animal studies in the introduction. Please also provide a few examples of animal studies, what the design was, the conclusion of those studies.

3. The manuscript should be finished with the clinical studies. This opinion review constantly mentions that there are conflicting data regarding vitamin D’s benefits in prostate cancer without giving any specific examples. Please explain in more detail the clinical trials by providing a few examples of the design and conclusions of those studies.

4. Figure 1 should be improved. Please add the following:

- 25(OH)D3 is the main circulating form.

- 1,25 (OH)2 D3 is an active metabolite, and it’s also called Calcitriol.

- Calcitriol induces epigenetic, transcriptome, and proteomic changes by interacting with VDR

Reviewer 3 Report

Comments and Suggestions for Authors

The authors should be congratulated on the topic proposed. However, the role of VitD in cancer is still assessed as irrelevant in several major studies (PMID: 30415629, 36495143). The review written does not add anything new to the actual literature. Despite this, the manuscript is well written. But it presents scant information (that needs to be addressed) on the molecular patterns involved in both PCA and VITD metabolism (PMID: 36235800). Maybe this aspect should be investigated and discussed. Moreover, the conclusion is weak. The authors should improve the introduction, discussion, and conclusion sections. A reconsideration is required.